# Missense Variants of von Willebrand Factor in the Background of COVID-19 Associated Coagulopathy

**DOI:** 10.3390/genes14030617

**Published:** 2023-02-28

**Authors:** Zsuzsanna Elek, Eszter Losoncz, Katalin Maricza, Zoltán Fülep, Zsófia Bánlaki, Réka Kovács-Nagy, Gergely Keszler, Zsolt Rónai

**Affiliations:** 1Institute of Biochemistry and Molecular Biology, Department of Molecular Biology, Semmelweis University, 1085 Budapest, Hungary; 2Department of Anesthesiology and Intensive Therapy, Bács-Kiskun County Teaching Hospital, 6000 Kecskemét, Hungary; 3Doctoral School, Semmelweis University, 1085 Budapest, Hungary

**Keywords:** COVID-19 associated coagulopathy, immunothrombosis, von Willebrand factor, ADAMTS13, single nucleotide polymorphism (SNP), missense variant, protein conformation

## Abstract

COVID-19 associated coagulopathy (CAC), characterized by endothelial dysfunction and hypercoagulability, evokes pulmonary immunothrombosis in advanced COVID-19 cases. Elevated von Willebrand factor (vWF) levels and reduced activities of the ADAMTS13 protease are common in CAC. Here, we aimed to determine whether common genetic variants of these proteins might be associated with COVID-19 severity and hemostatic parameters. A set of single nucleotide polymorphisms (SNPs) in the *vWF* (rs216311, rs216321, rs1063856, rs1800378, rs1800383) and *ADAMTS13* genes (rs2301612, rs28729234, rs34024143) were genotyped in 72 COVID-19 patients. Cross-sectional cohort analysis revealed no association of any polymorphism with disease severity. On the other hand, analysis of variance (ANOVA) uncovered associations with the following clinical parameters: (1) the rs216311 T allele with enhanced INR (international normalized ratio); (2) the rs1800383 C allele with elevated fibrinogen levels; and (3) the rs1063856 C allele with increased red blood cell count, hemoglobin, and creatinine levels. No association could be observed between the phenotypic data and the polymorphisms in the *ADAMTS13* gene. Importantly, in silico protein conformational analysis predicted that these missense variants would display global conformational alterations, which might affect the stability and plasma levels of vWF. Our results imply that missense vWF variants might modulate the thrombotic risk in COVID-19.

## 1. Introduction

The devastating COVID-19 pandemic caused by virulent strains of the severe acute respiratory syndrome coronavirus-2 (SARS-CoV-2) has overloaded healthcare systems worldwide with unprecedented morbidity, hospitalization, and mortality rates [1]. Critically ill patients were diagnosed with fulminant pneumonia presenting with massive inflammatory infiltrates that culminated in acute respiratory distress syndrome, requiring mechanical ventilation. Airspace filling and collapse due to inflammatory exudates and impaired oxygen diffusion on account of increased alveolar capillary permeability with consequent thickening of the alveolar wall resulted in intrapulmonary shunting, ventilation-perfusion mismatch, and severe hypoxemia, which is further aggravated by hypoperfusion of the lungs due to pulmonary thromboembolism [2]. The incidence of COVID-19 associated thrombosis was found to be as high as 31% among patients in intensive care units, the majority of which was venous thromboembolism [3]. Despite pre-hospital antiplatelet medication, thromboprophylaxis, and routine thrombolysis, autopsies revealed that 71.4% of patients having succumbed to the disease had thrombotic events with elevated D-dimer levels and prolonged prothrombin time [4].

The pathogenesis of SARS-CoV-2 infection associated pulmonary intravascular coagulopathy (PIC), also termed COVID-19 associated coagulopathy (CAC), has been characterized in detail [5,6]. COVID-19 leads to venous thromboembolism by establishing a generalized hypercoagulable state, with elevated plasma levels of key clotting factors including fibrinogen, factors V and VIII, as well as von Willebrand factor (vWF), which is a multidomain plasma protein essential for both platelet adhesion and aggregation [7]. One of the most sensitive molecular markers of CAC is a consistently elevated D-dimer concentration [8]. On the other hand, inflammatory cytokines cause local endothelial dysfunction in the lungs, resulting in thrombotic microangiopathy (immunothrombosis) [9]. A substantial prognostic factor of this condition is the elevation of the plasma vWF protein level, in combination with decreased ADAMTS13 enzyme activity [10]. ADAMTS13 is a liver-derived matrix metalloprotease known to attenuate the prothrombotic properties of vWF by cleaving it into two fragments [11]. In SARS-CoV-2 infection, pro-inflammatory cytokines, in particular IL-8 and TNF released from type 2 pneumocytes and alveolar macrophages, stimulate secretion of thrombogenic, unusually large vWF multimers (UL-vWFM) from the Weibel–Palade bodies of endothelial cells. On the other hand, several mechanisms have been reported to inhibit the production and/or to reduce the enzyme activity of ADAMTS13, including the overproduction of IL-6, IL-8, and TNF cytokines [12], the generation of anti-ADAMTS13 neutralizing antibodies [13], and the sequestration of vWF via binding to platelet factor 4 on the surface of activated thrombocytes [14]. UL-vWFM can bind to heparan sulfate in the glycocalyx of endothelial cells; this interaction activates the alternative complement pathway [15], which in turn, promotes the formation of tissue-factor rich neutrophil extracellular traps [16], mediators of pneumonia-associated pulmonary microvascular thrombosis [17]. Sustained endothelial dysfunction seems to be a major cytopathological determinant of long-term COVID-19 symptoms as well [18].

Genetic variations of the *vWF* and *ADAMTS13* genes were demonstrated to play a role in numerous pathological conditions related to thrombosis and hypercoagulability. Association analyses of complex traits usually aim to analyze common polymorphisms, although mutations and rare variants might also contribute to the genetic risk of these phenotypes [19,20]. Another element is the investigation of genetic variants with putative molecular effects. Missense polymorphisms modify the primary structure of the proteins and can result in altered function; on the other hand, SNPs, in either coding or non-coding sequences, can also have an impact on gene expression at numerous levels [21].

Secretion of vWF significantly increases in cases of impaired endothelium function (e.g., the damages of the endothelium caused by hypoxia, inflammation, or shear stress and high blood pressure). Taking this into account, the role of vWF develops in connection with thrombosis resulting from significant endothelial damage [22]. Polymorphisms of *vWF* might modulate the function of the protein, and these genetic variations were suggested to contribute to the risk of venous thrombosis. The rs1063856 SNP results in an amino acid change (p.Thr789Ala) in the D’ domain of vWF, playing a role in the interaction with FVIII. The modified vWF protein causes alterations in the function of FVIII [23], as well as increases the half-life of the protein [24,25], resulting in significantly increased plasma vWF-levels [26]. The rs216321 missense SNP modifies the primary structure of the D’D3 domain of vWF (p.Gln852Arg). This subunit also influences the FVIII level, as well as alters the interaction between the A1 domain and platelet Gp1b. The results are, however, controversial. The Gln coding allele was shown to cause decreased vWF levels and collagen induced platelet activation; on the other hand, the FVIII levels were increased. One explanation might be the alteration of the acid–base characteristics of the molecule, and thus, a local conformational change in the polypeptide chain, which might modify the binding site of FVIII [27]. The rs1800378 polymorphism was demonstrated to associate with pulmonary thromboembolism disease. The SNP results in an amino acid change in the D2 domain of vWF (p.His484Arg) [28], and interestingly, might cause a spurious increase in the plasma vWF level by modifying the antibody binding affinity of the protein [25], a phenomenon which might be of immunological significance. Two further missense polymorphisms (rs1800383 and rs216311) in the A1 interaction domain of vWF seem to have a synergistic effect. Interestingly, these SNPs have been shown to be connected with increased thrombosis risk, as well as with hemorrhagic events [29]. It was demonstrated that both SNPs can affect platelet aggregation by vWF with its cofactor ristocetin [30]; moreover, the rs216311 locus influenced interaction with FVIII [31]. The apparent contradictory effects might be resolved by the putative contribution of these SNPs to microangiopathy, which can be related to both systemic clinical conditions (i.e., thrombosis and hemorrhage).

The rs2301612 SNP (p.Gln448Glu) in the cysteine-rich domain of *ADAMTS13* plays an important role in substrate recognition and cleavage. Besides, the polymorphism is of significant interest because it is one of the most important targets of the autoimmune response in thrombotic thrombocytopenic purpura [32]. It was shown that this genetic variant may decrease the activity of the enzyme, resulting in increased vWF activity [33,34]. In addition, this polymorphism is associated with the risk of multivessel disease in a patient group with type 2 diabetes, raising the possibility that the locus might contribute to microthrombotic events in COVID-19 infection as well. The consequence of the rs34024143 SNP is an amino acid change in the signal peptide of the ADAMTS protein (p.Arg7Trp). This polymorphism might have an effect on the folding and stability of the mature protein, and signal sequence polymorphisms can influence the interaction with phospholipids and the components of the secretory pathway.

It was earlier demonstrated that the incidence of pulmonary thrombosis is significantly increased (but well below 100%) in critically ill COVID-19 patients [35]. Thus, it can be assumed that genetic factors might play a role in the development of the enhanced thrombotic risk. In light of the pivotal role of vWF and its processing peptidase ADAMTS13 in the development of COVID-19 associated coagulopathy, we aimed to explore whether certain missense polymorphisms of these genes can modulate (and predict) the severity and progression of COVID-19 by attenuating ADAMTS13 activities or stabilizing vWF.

## 2. Materials and Methods

### 2.1. Participants

A total of 72 study participants were recruited from in-patients testing positive for SARS-CoV-2 infection and treated at the Department of Emergency and/or Department of Anesthesiology and Intensive Care Unit of the Bács-Kiskun County Teaching Hospital, Kecskemét, Hungary. The diagnosis of SARS-CoV-2 infection was established using the standard RT-qPCR amplification protocol from nasopharyngeal swabs. COVID-19 pneumonia was confirmed by physical, laboratory, imaging, and microbiological examinations. Subjects with malignant, autoimmune, metabolic, or inflammatory comorbidities, as well as those receiving immunosuppressive therapy, were excluded from the analysis. The patients (*N* = 72) were grouped into severity cohorts, according to WHO criteria. All participants were of Caucasian origin.

### 2.2. DNA Sampling and Purification

Sample collection was carried out using Tempus™ Blood RNA Tubes obtained from ThermoFisher. Samples were stored at room temperature in the hospital for a maximum of 5 days and subsequently transported to the Molecular Genetic Laboratory (Department of Molecular Biology, Semmelweis University) and were kept at −20 °C until further processing.

DNA isolation was initiated by adding 450 µL cell proteinase K buffer (0.1 M NaCl, 0.01 M Tris-HCl pH = 8, 0.5% SDS, 0.2 mg/mL proteinase K) to 800 µL of the sample, followed by incubation at 56 °C overnight. Proteins were then precipitated using saturated NaCl and removed by centrifugation. DNA was isolated from the supernatant using the standard ethanol/isopropanol precipitation method. Precipitated DNA was redissolved in 0.5× TE (0.005 M Tris-HCl, pH = 8 and 0.5 mM EDTA). DNA concentrations were measured with a Nanodrop1000 spectrophotometer. The average DNA concentration of the samples was 140 ng/µL (20–572 ng/µL).

### 2.3. In Silico Tools for SNP Selection and Genotype Analysis

Genomic sequences of the *ADAMTS13* and *vWF* genes, as well as data about the SNPs, were obtained from NCBI Genebank (http://www.ncbi.nlm.nih.gov/gene, accessed on: 19 July 2022) and Ensembl (http://www.ensembl.org/index.html, accessed on: 19 July 2022). Allele frequency data were obtained from the gnomAD database (https://gnomad.broadinstitute.org/, accessed on: 19 July 2022). To focus on the analysis of SNPs with putative functional effects, missense SNPs were predominantly selected for further investigation.

Restriction endonucleases for PCR–RFLP genotyping were selected with help of the NEBcutter v2.0 tool (http://nc2.neb.com/NEBcutter2/, accessed on: 26 July 2022). Primers were designed using the NCBI Primer-Blast software (https://www.ncbi.nlm.nih.gov/tools/primer-blast/, accessed on: 26 July 2022) and checked with the Oligonucleotide Properties Calculator (http://biotools.nubic.northwestern.edu/OligoCalc.html, accessed on: 26 July 2022).

### 2.4. SNP Genotyping

#### 2.4.1. Genotype Analysis with qPCR TaqMan Probes

Predesigned SNP genotyping assays, containing 2 primers to amplify the adjacent region of the polymorphisms and 2 allele-specific probes labeled with FAM and VIC fluorescent dyes, respectively, were obtained from ThermoFisher (Waltham, MA, USA). Assay IDs were as follows: SNPs in *ADAMTS13*: rs2301612: C__11571465_1_, rs34024143: C__3183339_10; SNPs in *vWF*: rs1800378: C__8921130_20, rs1800383: C__11915632_10, rs216311: C_175678231_10, and rs1063856: C__3288406_30. The reaction mixtures contained 1× PCR TaqMan ProAmp mastex mix (AmpliTaq Gold^®^ DNA polymerase, dNTPs, ROX dye, MgCl_2_ and buffer), 1× TaqMan probe, and 4 ng genomic DNA in a final volume of 6 µL. Thermocycles were initiated with a 95 °C—10 min step to activate the hot-start AmpliTaq Gold^®^ DNA polymerase, followed by 40 cycles of 95 °C—15 s denaturation and 60 °C—1 min combined annealing and extension. A reporter signal was detected during this latter step, and an endpoint allelic discrimination analysis was also performed after PCR amplification to classify the samples into three clusters, according to their genotype.

#### 2.4.2. PCR–RFLP

Three SNPs (*ADAMTS13:* rs2301612, rs28729234, and *vWF:* rs216321) were analyzed by PCR–RFLP. To boost the reliability of the genotyping, primers were designed to include a non-polymorphic, control digestion site into the PCR amplicon in the case of each polymorphism. Table 1 summarizes the technical parameters of the PCR–RFLP methods.

PCR amplification was carried out using 0.25 U HotStarTaq DNA polymerase (Qiagen, Hilden, Germany) dissolved in 1× buffer and 1× solution Q. The reaction mixtures contained 0.2 mM of each deoxyribonucleoside triphosphate (dATP, dCTP, dGTP, and dTTP), 1 µM sense and antisense primers, and 4 ng genomic DNA in a total volume of 10 µL. Thermocycles were initiated by denaturation and polymerase activation at 95 °C for 15 min, followed by 40 cycles of denaturation at 94 °C for 30 s, annealing at 60 °C for 30 s, and extension at 72 °C for 1 min, terminated by a final extension step at 72 °C for 10 min. RFLP analysis was carried out by adding 6 µL of a restriction endonuclease mixture containing 0.2 U restriction endonuclease in 1× CutSmart Buffer (New England Biolabs, Ipswich, MA, USA) to the PCR-amplicons, and samples were incubated at 37 °C for 3 h. The digestion products were resolved by conventional slab gel electrophoresis (2.5% agarose, 13 V/cm electric field). After staining in 0.5 µg/mL ethidium bromide solution for 10 min, the migration patterns were imaged using a GelDoc 1000 gel documentation system (BioRad, Hercules, CA, USA).

### 2.5. Prediction of 3D Protein Structure

The I-Tasser software (https://zhanggroup.org/I-TASSER/about.html, accessed on: 22 August 2022) was used to predict the effect of missense SNPs on the corresponding protein conformation. The 3D structure of each haplotype was plotted using the Swiss-PdbViewer 4.1.0 (https://spdbv.unil.ch, 22 August 2022).

### 2.6. Statistical Analysis

The Hardy–Weinberg equilibrium (HWE) for genotype distributions was analyzed with the *χ*^2^-test. Association analyses were carried out by comparing the genotype frequencies of each polymorphism in all patient cohorts using SPSS v17.0 and HaploView v4.2 [36]. The Bonferroni correction for multiple testing was used to rule out false positive results in the cohort analysis. As 8 SNPs were investigated, the modified level of statistical significance was *p* < 0.00625. Linkage disequilibrium and haplotype analyses were performed by HaploView v4.2, and correction for multiple testing was conducted by permutation analysis (100,000 random permutations). Genotype data of a healthy, Caucasian population for linkage disequilibrium testing were obtained from the 1000 Genomes Browser (https://www.internationalgenome.org/1000-genomes-browsers/index.html, accessed on: 14 August 2022). The analysis of variance (ANOVA) procedure was used to seek association between clinical parameters as continuous variables and genotype data (version 3.05).

## 3. Results

### 3.1. Clinical Characterization of Patient Cohorts

Clinical progression scores were assigned to all participants according to the clinical progressions scores of WHO [37]. Four cohorts were created, comprising patients with scores of 4, 5, 6–7, and 8–9, respectively, presenting with increasingly severe COVID-19 symptoms (Table 2). The subjects were predominantly males, with a mean age around 50 years in each cohort. Red blood cell counts and hemoglobin levels did not differ significantly across the cohorts, but creatinine concentrations were slightly elevated in cohorts 5 to 8–9. The duration of hospitalization and ICU treatment, the number of infiltrated lobes (lobar involvement), respiratory rates, and Horowitz coefficients (a measure of alveolar gas transfer efficiency) were proportional to the severity of the disease. Importantly, plasma levels of interleukin-6, a pleiotropic pro-inflammatory cytokine known to induce synthesis of acute phase proteins in hepatocytes [38], correlated well with those of its target genes CRP and fibrinogen, with dramatic upregulation in cohort 5 and beyond, although a marked drop was also seen comparing 6–7 to 8–9. Plasma levels of D-dimer, a widely used thrombosis marker in COVID-19 [39], increased parallel with the severity scores, despite essentially unchanged INR values.

### 3.2. SNP Genotyping

As both vWF and ADAMTS13 have been shown to play a crucial role in the development of thromboembolism in COVID-19, we aimed to explore whether their missense variants might be correlated with progression scores of the disease by directly modulating the prothrombotic activity of these proteins. To test this hypothesis, a set of SNPs were genotyped in both genes, and association analyses were conducted to uncover any statistically significant enrichment of a certain allele in patient cohorts.

The selection criteria of missense SNPs were as follows. We aimed to investigate genetic variants with MAF > 5% in the Caucasian population. We preferred polymorphisms with putative biological effect, such as missense SNPs, where the two-allele codes for amino acids greatly different in terms of charge, size, or hydropathy index [40]. We checked the availability of literary data in connection with the functional role of the SNPs in terms of modulating thrombotic risk or protein plasma levels, etc. According to these criteria, five SNPs from the *vWF* and three from the *ADAMTS13* genes were selected for analysis (Table 3).

Two missense SNPs in the *vWF* gene (rs1063856 and rs216321) caused amino acid changes in the D’ multimerization domain of the protein. These polymorphisms were shown to be associated with plasma vWF:Ag levels [26]. On the other hand, the TT variant of the rs1800378 SNP was positively correlated with the incidence of pulmonary thromboembolism [28]. The rs216311 SNP was found to affect vWF:Ag plasma levels in a blood group dependent manner [41] and, along with rs1800383, was linked to bleeding history in a Nigerian population [29]. Both SNPs alter the primary sequence of the A1 domain that is known to interact with both the platelet integrin glycoprotein Ib and subendothelial collagen.

The rs2301612 SNP in the ADAMTS13 gene is associated with cerebral aneurysms, possibly via altered arterial wall remodeling [42]. However, no functional information was available on the rs34024143 missense and the intronic rs28729234 polymorphisms.

SNPs were genotyped either by real-time PCR or PCR-RFLP, as described in Section 2.3. Genotyping of the rs2301612 SNP was performed, with both methods producing identical results, proving the reliability of our genotyping protocols. Genotype frequencies are presented in Table 4. Hardy-Weinberg equilibria were fulfilled for each SNP as no significant difference (*p* > 0.1) could be observed between the measured and expected genotype frequencies for any of the investigated SNPs. In case of two ADAMTS13 SNPs, however, minor allele frequencies (MAFs) fell slightly outside the range of 5–40% declared as inclusion criterion (4% for rs28729234 and 44% for rs2301612). This deviation from the gnomAD genotype data might be attributed to the low sample numbers in our study. Furthermore, no minor allele homozygotes were found in the case of rs1800383 and rs28729234, likely for the same reason.

### 3.3. Linkage Disequilibrium and Haplotype Analysis

Figure 1A represents Lewontin’s *D’* and *R*^2^ values from the pairwise linkage disequilibrium (LD) analysis in our patient population. These results are in good agreement with those obtained from the LD analysis using the dataset of the 1000 Genomes Project (Figure 1B). Generally high levels of LD can be observed in the ADAMTS13 gene region. It is notable that the rs1800378 polymorphism in vWF was not in linkage disequilibrium with the other sites, in agreement with the fact that it is relatively far from the other SNPs on the chromosome. On the other hand, high *D’* values, in combination with low *R*^2^ values in the context of the rs1800383 polymorphism, disclose high-level linkage disequilibrium of this locus with other SNPs. The unusually high (100% and 90%) D’ scores between this and the rs216311 and rs216321 SNPs might imply the existence of a haplotype block encompassing these relatively close SNPs. 

### 3.4. Association Analyses

In an attempt to determine whether genetic polymorphisms in the *vWF* and *ADAMTS13* genes are associated with the progressive severity of COVID-19, *χ*^2^-tests were performed by comparing the genotype frequencies of each polymorphism in all patient cohorts. As it shown by the data presented in Table 5, no correlation could be detected between the SNPs and the clinical progression scores (i.e., *p* ≥ 0.15 for all polymorphisms, higher than that of both the nominal (*p* > 0.05), as well as the corrected level (*p* > 0.00625) of statistical significance) in this population. As no complete linkage disequilibrium was observed between the investigated loci, haplotypes were also constructed, and their frequencies (Appendix A) were compared in the four cohorts, nor could statistically significant differences be found by employing this approach. However, considering clinical parameters as continuous variances across all cohorts and implementing the ANOVA approach, it turned out that the rs1800383 and rs216311 SNPs affecting the A1 domain of vWF were significantly associated with the hemostatic parameters. Table 6 summarizes data with significant associations; further results of the ANOVA analyses related to each clinical parameter are shown in Appendix A. Patients carrying the rs1800383 CC homozygote genotype showed average fibrinogen levels of 6.64 g/L. In contrast, mean fibrinogen levels in the CG heterozygotes were 17.62 g/L (*p* = 0.032). On the other hand, the T allele of the rs216311 SNP seems to prolong the tissue factor dependent clotting time, as indicated by greater international normalized ratios (INR) (*p* = 0.039).

Though seemingly less relevant in terms of pulmonary thromboembolism, the minor TT genotype of the rs1063856 SNP was found to be associated with significantly lower red blood cell counts, as well as lower hemoglobin and creatinine levels, in our patients with nominal *p* values of 0.049, 0.029, and 0.008, respectively (Table 6).

The rest of the SNPs in the vWF and ADAMTS13 genes did not show association with any clinical parameter.

### 3.5. Predicted Structural Changes in Missense Variants

In light of the above associations, we aimed to explore whether the missense variants of vWF and ADAMTS13 have any effect on protein conformation, using the ResQ B factor profiling tool [43]. This method predicts the effects of amino acid exchanges on the local secondary structure, solvent accessibility, residue-level accuracy, and thermal mobility (B-factor), while the I-Tasser software was employed to build global conformational models of the polypeptide. Results of in silico conformational analyses are presented in Table 7. Of the 7 allelic variants analyzed, only the rs2301612 G allele triggered a local conformational change, the disruption of an α helical structure by replacing glutamine with glutamate in the ADAMTS13 protein. The T allele of rs216321, resulting in an arginine-to-glutamine exchange, substantially altered solvent accessibility, as glutamine is no longer exposed to the surface of the protein (‘E’ to ‘B’ transition).

The only minor variant that confers substantial stabilization on the corresponding protein is the G allele of the rs1800383, encoding an aspartate residue with a negative predicted normalized B factor (BFP) of −0.11. In comparison, the major allele (C) incorporates histidine in the same position, which is slightly less stable (BFP = 0.04). Importantly, the residue-specific quality (RSQ) scores corresponding to the estimated deviation of the residue from the native structure of the protein exhibit the greatest difference in the context of this polymorphism, when compared to the others (4.83 vs. 6.94, a more than 40% increase from the more frequent His to the rare Asp-containing version). Based on these data, it seemed reasonable to assume that the His-to-Asp change might induce not only a local, but even a more extensive structural change in the vWF.

To validate this notion, structural models of both proteins with just single amino acid exchanges were generated, as displayed in Figure 2. Interestingly, not only the Asp1472, but also the Thr1381 and Ala789, containing variants of vWF, exhibited dramatic predicted conformational changes at the global level. The Asp1472 protein seems to be less compact, with several disrupted secondary structures, while a hallmark of Thr1381 and Ala789 versions is a large unstructured domain, either jutting out from the central body of the protein (Thr1381), or connecting its N- and C-terminal regions (Ala789). It is remarkable to note that substitution of Arg484 with His and Arg852 with Gln does not lead to conformational changes as extensive as those seen in the case of any of the three other isoforms. In summary, it is remarkable that three minor isoforms of vWF, which are associated with elevated Fg levels and INR values, as well as increased RBC counts, plasma Hb, and creatinine concentrations, have steric structures quite different from that of the most frequently occurring haplotype.

Although not showing any association with clinical parameters, structural models of the ADAMTS13 variants were also created. Surprisingly, they display more stretched conformations which are much different from the rather compact structure of the most frequently seen haplotype (Figure 2, upper panel).

## 4. Discussion

Over the past two and one-half years, a plethora of scientific information has been published on the pathomechanism of COVID-19. Despite being primarily a respiratory infection, an extraordinarily high incidence of thrombotic events has been observed in severe cases, leading investigators to define a new entity termed, ’COVID-19 associated coagulopathy’ (CAC). CAC is derived from an inflammation-related imbalance of elevated plasma vWF levels and decreased ADAMTS13 activities, presumably mediated by inflammatory cytokines and complement factors. Importantly, clarification of the pathogenesis incited clinical trials with targeted therapeutic modalities, including administration of caplacizumab, an anti-vWF monoclonal antibody [44], activated complement C5a inhibitors [45], recombinant ADAMTS13 [46], and IL1 és IL6 antibodies [47].

Just as is the case with all infectious diseases, the clinical course of SARS-CoV-2 infections is determined not only by virulence factors of the invading pathogen, but also by genetic factors of the host, which govern innate and adaptive immune responses [48]. To obtain an insight into individual differences in the defense mechanisms of the host, high-throughput, genome-wide association studies were conducted to map COVID-19-specific susceptibility loci in the human genome [49,50]. While hemostatic abnormalities are proven to contribute to the mortality of COVID-19 to a large extent, genetic polymorphisms modulating the thrombotic risk of this disease have scarcely been investigated. Recently, a systems biology approach was described by Abu-Farha et al. [51] to identify high-risk patients, and a phenome-wide association study was performed to uncover the genetic susceptibility for COVID-19-associated thrombosis [52]. Lapic et al. [53] analyzed the role of various missense variants of coagulation factors II, V, and XIII, as well as further proteins supposed to modulate thrombotic risk in COVID-19, including platelet integrin receptors, the endothelial nitric oxide synthase, and methylene tetrahydrofolate reductase. However, to the best of our knowledge, ours is the first study assessing the potential contribution of a set of single nucleotide polymorphisms in the vWF and ADAMTS13 genes, key factors in the background of CAC, to increased thrombotic risk among COVID-19 patients.

In the present study, COVID-19 patients were stratified in severity cohorts using internationally acknowledged progression scores [37], and association analyses were performed between the above mentioned SNPs and disease progression, but no statistically significant correlations were found in this context. Either our study was underpowered due to low sample size, or single missense effects are not powerful enough to significantly modify the progression of the disease. Therefore, we proceeded to seek more specific associations between the same SNPs and laboratory findings of patients in general, and hemostatic parameters, in particular. Implementing this approach, three missense SNPs in the vWF gene were found to associate with fibrinogen levels, clotting time (INR), and red blood cell, hemoglobin and creatinine levels in COVID-19 patients, but we failed to find any correlations with *ADAMTS13* SNPs.

As far as ADAMTS13 is concerned, surprisingly little is known about the association of its genetic variants with any disease. To date, only two studies have been published in this context, the one concluding that the A allele of the intronic rs4962153 SNP confers protection from cerebral malaria [54]. The other investigation found that the rs2301612 SNP, known to be a key determinant of plasma ADAMTS13 levels [33], is associated with cerebral aneurysms [42] due to abnormal arterial wall remodeling. The latter polymorphism was included in our study as well, but neither this nor the two other SNPs were found to be associated with disease severity or laboratory parameters in COVID-19 patients (Table 5 and Table 6). However, our in silico structure analysis revealed that tertiary structures of missense Gln448Glu variants are markedly different (Figure 2), an interesting finding that might underlie differences in plasma levels, rate of secretion, and specific enzyme activities between the Gln448 and Glu448 containing common ADAMTS13 variants [33]. Importantly, the common variant containing Trp in position 7 is predicted to display conformational alterations strikingly similar to those elicited by a Gln448Glu exchange (Figure 2). This observation might also prompt in vitro studies exploring the stability and activity of this common variant.

The literature on the genetic associations of missense variants of *vWF* with hemostatic abnormalities is far more extensive. Importantly, every single SNP we found to be associated with hematological parameters (rs1800383, rs216311, and 1063856, but not the other two) has been shown to modulate the plasma levels of the vWF protein [26,41]. Structural modeling revealed that only these missense variants (but not the rest) entail considerable conformational changes which might also influence the stability and thereby, the plasma levels, of the protein (Figure 2).

Both vWF and fibrinogen serve as molecular adhesives during the formation of platelet-rich thrombi, and both proteins are substrates of plasmin, a fibrinolytic protease. It has been shown that the rate of fibrinogen degradation by plasmin is significantly slower in the presence of vWF [55]. This correlation might be implicated in the differential regulation of fibrinogen levels by missense variants of vWF, resulting in higher Fg levels in rs1800383 CG heterozygotes in our study. One can assume that Fg levels would be even higher in individuals harboring the GG genotype, but such a genotype did not occur in our patient cohorts, unfortunately.

Cleavage of vWF multimers by plasmin under shear stress has recently been shown to compromise platelet adhesion to collagen [56]. This mechanism might offer a plausible explanation for the prolongation of tissue thromboplastin-induced clotting time (prothrombin time) and consequently elevated INR in rs216311 TT homozygotes. Namely, the Thr1381 containing the vWF variant might be a better substrate of plasmin, resulting in lower circulating vWF levels, impaired platelet adhesion, aggregation, and secondary hemostasis.

Finally, a relatively old observation might provide a plausible clue for the seemingly elusive association of the minor C allele of the rs1063856 polymorphism, with slightly elevated red blood cell counts and hemoglobin levels. Wick et al. reported that unusually large von Willebrand factor multimers help sequester young healthy erythrocytes on endothelial cells [57]. It is tempting to speculate that the Ala789-containing variant is less potent in doing so, leaving more red blood cells suspended in the circulation. On the other hand, higher creatinine levels measured in the presence of the Ala789 variant might simply reflect impaired renal function due to thrombotic hypoperfusion of the kidneys.

The major limitation of the cohort analysis portion of the presented work is the rather low sample size. Thus, it can be considered as a pilot study only; however, the predicted effects of missense variants of vWF on protein conformation and biological activity deserve further research. The putative, weak associations observed here should be verified in further studies investigating cohorts including a much higher number of participants. Another limitation is the lack of ADAMTS13 and vWF level measurements and functional characterization of the proteins possessing different allelic and haplotype variants. These analyses could provide further insights in the biological role of the investigated SNPs, both at molecular and clinical levels.

In conclusion, the common SNPs investigated in our study were predicted to result in substantial conformational changes in the vWF protein, potentially modifying its biological activity. Our small cohort analysis did not reveal an association between the selected polymorphisms and the severity of COVID-19 infection; on the other hand, we found a connection between three missense variants of the vWF gene and some clinical variables (INR, fibrinogen, creatinine, hemoglobin levels, and red blood cell counts). Importantly, these findings can be considered as pilot results only, and further association studies, as well as functional characterization of the polymorphisms, are required to shed light on the exact role of these missense SNPs.

## Figures and Tables

**Figure 1 genes-14-00617-f001:**
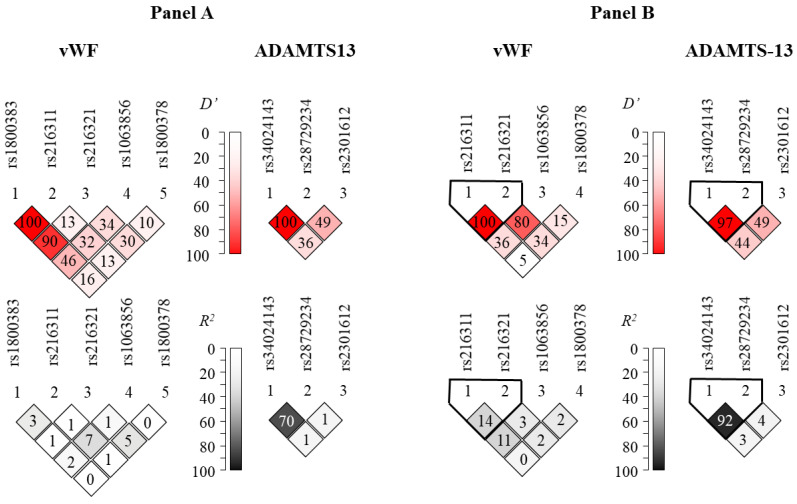
Representation of pairwise linkage disequilibrium analysis of SNPs genotyped in the *vWF* (**left column**) and *ADAMTS13* (**right column**) genes (**Panel A**). **Panel B** represents linkage disequilibrium data calculated from datasets of the 1000 Genomes Project (https://www.internationalgenome.org, accessed on 22 August 2022). Note that no allele frequency data were available for the rs1800383 in the latter database. Color-coded Lewontin’s *D*’ and *R*^2^ values uncover haploblocks with relatively low recombination frequencies.

**Figure 2 genes-14-00617-f002:**
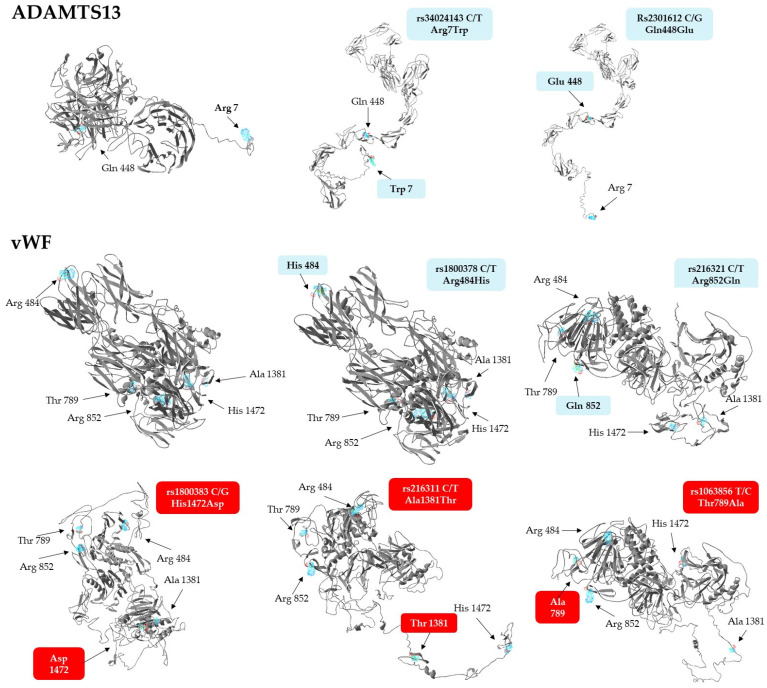
Steric structures of vWF and ADAMTS13 missense variants, as predicted by the I-Tasser software. Images were generated by exchanging single amino acids encoded by the minor allele of each genotyped SNP. Variants of vWF found to be associated with laboratory parameters are highlighted against a red background.

**Table 1 genes-14-00617-t001:** PCR–RFLP based genotyping of the ADAMTS13 rs2301612, rs28729234, and vWF rs216321 SNPs. “^” indicates the cleavage site in the recognition sequence of the restriction endonucleases. SNPs are printed in bold. Lengths of digestion products shown in bold are the allele-specific fragments; normal typesetting refers to control products.

Polymorphism	rs2301612	rs28729234	rs216321
primers	5′ CCCGGGGTTTTCCCATCAA 3′ 5′ TCAGAGATGGGATGTCAGTGC 3′	5′ CTCACAAAAGGCCACGCTTC 3′ 5′ AGCAGGTTCTCACCATGCAC 3′	5′ CCTACCGCTTCAGGCACTTC 3′ 5′ CACACTCCACGCTACAGGTC 3′
length of PCR product (bp)	520	370	457
restriction endonuclease	*Hpy*88 I (TCN^**G**A)	*Hpa* II (C^**C**GG)	*Nla* IV (GGN^NC**C**)
length of digestion products (bp)	C: **378**, 140, 2 G: **231**, **147**, 140, 2	C: 65, 12, 11, 11, 60, 20, **108**, **83** T: 65, 12, 11, 11, 60, 20, **191**	C: **46**, **291**, 56, 47, 17 T: **337**, 56, 47, 17

**Table 2 genes-14-00617-t002:** Clinical and biological characterization of subject cohorts. Participants of the study were ordered into severity cohorts according to WHO clinical progression scores [37]. Subjects with scores of 6 or 7 and 8 or 9 were classified in a common cohort, respectively, in order to guarantee that all cohorts included at least ten members for sufficient statistical significance. Lobar involvement corresponds to the number of infiltrated lung lobes. Standard deviations are shown in brackets.

WHO Clinical Progression Score	4 (*n* = 10)	5 (*n* = 35)	6 or 7 (*n* = 10)	8 or 9 (*n* = 17)
ratio of males (%)	70	69	70	88
mean age (years)	50.6 (12.5)	49.1 (11.3)	52.8 (10.1)	49.8 (8.5)
body mass index (BMI) (kg/m^2^)	26.5 (2.7)	30.1 (10.1)	28.6 (2.7)	33.1 (7.1)
body mass index (BMI) > 30 (%)	25	37	40	57
red blood cell count (T/L)	4.6 (0.5)	4.7 (0.5)	4.9 (0.6)	4.7 (0.6)
hemoglobin (g/L)	139.6 (17.2)	140.1 (14.7)	142.9 (16.6)	142.8 (18)
creatinine (µM)	77.1 (30.7)	89.6 (35.4)	92.3 (28.7)	91.4 (35)
treatment in hospital (days)	9.2 (6.1)	10.6 (4.2)	19.5 (10.5)	24.5 (23.3)
treatment in intensive care unit (days)	0.2 (0.6)	0.3 (0.8)	4.1 (5.5)	12.9 (8.6)
lobar involvement	1.6 (1.9)	3.2 (1.9)	4.8 (0.5)	4.4 (1.1)
respiratory rate (1/min)	16.2 (2.3)	19.9 (5.6)	28.5 (5.2)	34.5 (6.0)
Horowitz coefficient (PaO_2_/FiO_2_ Hgmm)	–	150.5 (100.6)	100.3 (74.2)	61.2 (14.5)
interleukin-6 (IL-6) (pg/mL)	11.1 (12.3)	85.3 (158.9)	83 (149.8)	64.4 (60.2)
C reactive protein (CRP) (mg/L)	75.3 (69.9)	149 (99)	168 (107.2)	153.5 (105)
fibrinogen (g/L)	5.4 (0.2)	7.5 (1.2)	7.7 (1.6)	6.9 (2.5)
D-dimer (mg/L)	3.4 (7.2)	1.8 (3.5)	5.2 (9.2)	9.6 (18.3)
international normalized ratio (INR)	1.3 (0.2)	1.2 (0.1)	1.3 (0.3)	1.3 (0.2)
hypertension (%)	33	44	33	38
alteplase treatment (%)	0	0	33	62
noradrenalin treatment (%)	0	3	0	85

**Table 3 genes-14-00617-t003:** Data of SNPs involved in our study. Genomic locations of SNPs are shown according to the Genome Reference Consortium Human Build 37 patch release 13 (GRCh37.p13). Alleles and the corresponding amino acids in the encoded protein variants are shown in the ’major/minor’ order, as obtained from the gnomAD database. rs: Reference SNP cluster ID.

Gene	SNP	Allele	Genomic Location	Type of Polymorphism	Amino Acid Exchange	Protein Domain Affected
vWF	rs1800383	C/G	12:6,128,170	missense	His1472Asp	A1
vWF	rs216311	C/T	12:6,128,443	missense	Ala1381Thr	A1
vWF	rs216321	C/T	12:6,143,984	missense	Ala852Gln	D’
vWF	rs1063856	T/C	12:6,153,534	missense	Thr789Ala	D’
vWF	rs1800378	C/T	12:6,172,202	missense	Arg484His	D2
ADAMTS13	rs34024143	C/T	9:136,287,582	missense	Arg7Trp	S
ADAMTS13	rs28729234	C/T	9:136,295,232	intronic	–	–
ADAMTS13	rs2301612	C/G	9:136,301,982	missense	Gln448Glu	Cys-rich

**Table 4 genes-14-00617-t004:** Genotype distribution of analyzed SNPs across all cohorts. MAF: minor allele frequency. Levels of statistical significance for Hardy–Weinberg equilibrium (HWE) were calculated using the *χ*^2^-test.

	Major Allele Homozygote	Heterozygote	Minor Allele Homozygote	HWE (*p*)	MAF	Genotyping Method
**rs1800383**	0.89 (CC)	0.11 (CG)	0.00 (GG)	0.87	5% G	TaqMan
**rs216311**	0.37 (CC)	0.48 (CT)	0.15 (TT)	0.99	39% T	TaqMan
**rs216321**	0.53 (CC)	0.46 (CT)	0.01 (TT)	0.11	24% T	PCR-RFLP
**rs1063856**	0.50 (TT)	0.35 (CT)	0.15 (CC)	0.23	32% C	TaqMan
**rs1800378**	0.40 (CC)	0.48 (CT)	0.12 (TT)	0.94	36% T	TaqMan
**rs34024143**	0.91 (CC)	0.08 (CT)	0.01 (TT)	0.20	5% T	TaqMan
**rs28729234**	0.93 (CC)	0.07 (CT)	0.00 (TT)	0.96	4% T	PCR-RFLP
**rs2301612**	0.34 (CC)	0.45 (CG)	0.21 (GG)	0.78	44% G	both

**Table 5 genes-14-00617-t005:** Genotype frequencies of minor allele homozygotes in disease severity cohorts. Statistical analysis was performed with SPSS v17.0 software.

SNP	MAF Genotype	Genotype Frequencies in Cohorts				*p* Value
		4	5	6–7	8–9	
**rs1800383**	GG	0.00	0.00	0.00	0.00	0.62
**rs216311**	TT	0.06	0.24	0.22	0.00	0.15
**rs216321**	TT	0.00	0.03	0.00	0.00	0.67
**rs1063856**	CC	0.13	0.18	0.11	0.13	0.96
**rs1800378**	TT	0.13	0.09	0.22	0.13	0.89
**rs34024143**	TT	0.06	0.00	0.00	0.00	0.54
**rs28729234**	TT	0.06	0.00	0.00	0.00	0.59
**rs2301612**	GG	0.06	0.26	0.11	0.25	0.61

**Table 6 genes-14-00617-t006:** Genetic association of certain laboratory parameters with polymorphisms. **Panel A.** *p* values were calculated by ANOVA. Nominally significant (*p* < 0.05) results are highlighted using a gray background. Only parameters with significant association are shown in the table. RBC: red blood cell count (T/L); INR: international normalized ratio. Hemoglobin and fibrinogen levels are shown in g/L, whereas creatinine concentrations are measured in µmol/L. **Panel B.** Genotype–phenotype correlations of significant associations indicated in panel (A). Average levels and standard deviations (enclosed in brackets) of clinical parameters are shown in the context of each genotype group.

**Panel A**										
**Clinical** **Parameter**	**ADAMTS13 SNPs**			**vWF SNPs**						
	**rs2301612**	**rs28729234**	**rs34024143**	**rs1800383**		**rs216311**		**rs216321**	**rs1063856**	**rs1800378**
**RBC**	0.728	0.754	0.853	0.545		0.667		0.610	**0.049**	0.086
**hemoglobin**	0.646	0.692	0.652	0.625		0.543		0.358	**0.029**	0.060
**INR**	0.505	1.000	1.000	0.779		**0.039**		0.976	0.686	0.570
**fibrinogen**	0.461	1.000	1.000	**0.032**		0.441		0.917	0.836	0.332
**creatinine**	0.753	0.933	0.927	0.549		0.976		0.958	**0.008**	0.544
**Panel B**										
**SNP**	**clinical** **parameter**	**major homozygote**			**heterozygote**			**minor homozygote**		
**rs1063856**	RBC (T/L)	TT	4.59	(2.33)	CT	4.89	(2.17)	CC	5.00	(1.86)
**rs1063856**	hemoglobin (g/L)	TT	134.94	(68.36)	CT	145.00	(64.45)	CC	147.18	(54.82)
**rs216311**	INR	CC	1.20	(0.41)	CT	1.23	(0.50)	TT	1.45	(0.38)
**rs1800383**	fibrinogen (g/L)	CC	6.64	(3.23)	CG	17.62	(4.71)	GG	–	–
**rs1063856**	creatinine (µM)	TT	84.78	(46.52)	CT	71.45	(35.38)	CC	111.00	(44.73)

**Table 7 genes-14-00617-t007:** Conformational analysis of missense variants of vWF and ADAMTS13. Quantitative descriptors of predicted steric structures are shown in relation to the most frequently occurring haplotype (‘R484 T789 R852 A1381 H1472’ for vWF (left panel) and ‘R7 Q448’ for ADAMTS13 (right panel), respectively). The descriptors are as follows: SS, predicted secondary structure; SA, predicted solvent accessibility; COV, threading alignment coverage; BFP, predicted normalized B-factor; RSQ, residue-specific quality.

vWF						ADAMTS13					
Haplotype	SS	SA	COV	BFP	RSQ	Haplotype	SS	SA	COV	BFP	RSQ
**Arg484His**						**Arg7Trp**					
nt: **C**TCCC aa: **R**TRAH	S	B	0.62	−0.47	3.80	nt: **C**C aa: **R**Q	C	E	0.10	0.16	28.16
nt: **T**TCCC aa: **H**TRAH	S	B	0.62	−0.49	3.88	nt: **T**C aa: **W**Q	C	E	0.09	0.11	33.08
**Thr789Ala**						**Gln448Glu**					
nt: C**T**CCC aa: R**T**RAH	C	B	0.39	−0.26	4.17	nt: C**C** aa: R**Q**	H	B	0.57	−0.43	3.74
nt: C**C**CCC aa: R**A**RAH	C	B	0.41	−0.21	4.14	nt: C**G** aa: R**E**	C	B	0.63	−0.28	4.72
**Arg852Gln**											
nt: CT**C**CC aa: RT**R**AH	C	E	0.56	−0.02	4.18						
nt: CT**T**CC aa: RT**Q**AH	C	B	0.55	−0.13	3.92						
**Ala1381Thr**											
nt: CTC**C**C aa: RTR**A**H	S	B	0.28	−0.94	4.34						
nt: CTC**T**C aa: RTR**T**H	S	B	0.26	−0.88	4.25						
**His1472Asp**											
nt: CTCC**C** aa: RTRA**H**	C	E	0.23	0.04	4.83						
nt: CTCC**G** aa: RTRA**D**	C	E	0.19	−0.11	6.94						

## Data Availability

Individual clinical and genetic data are available upon request sent directly to the corresponding author.

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
