# Peer review of "Missense Variants of von Willebrand Factor in the Background of COVID-19 Associated Coagulopathy"

_genes, 2023, doi:10.3390/genes14030617_

Round 1

Reviewer 1 Report

Genes-2183443

This is a very small candidate gene association study in which the authors genotyped five SNPs in the gene for von Willebrand factor (vWF) and two SNPs in the gene for ADAMTS13 in 72 individuals diagnosed with Covid-19. The manuscript is well written, but there is a huge problem with the study and that is the sample size: only 72 individuals were included in the study. Many authors have pointed out in the past that the minimum sample size for any genetic association study is 250. Results from studies smaller than that are often false positives or false negatives and cannot be replicated in any subsequent study. Most studies now have thousands if not millions of specimens to obtain stable estimates of the genetic effects under study. We must continue to make sure that published studies meet the minimum sample size threshold.

The authors state that they used Bonferroni correction for multiple testing, but do not provide any details on how this was done. It would be important to provide both raw p-values and the corrected p-values and indicate how many conditions, models and SNPs were tested for each result.

Reviewer 2 Report

In this manuscript, Zsuzsanna Elek and other authors discovered a set of single nucleotide polymorphisms in the vWF and ADAMTS13 genes associated with some of the clinical hemostatic parameters but not associated with Covid-19 disease severity.

The introduction is relevant and sufficient background information about Covid-19 associated coagulopathy as well as clinical manifestation is presented. There are only three issues about the experiment design and data presentation.

1.    In line 74-76, you should provide reference article to support the claim that the incidence of pulmonary thrombosis is higher in critically ill Covid-19 patients.

2.    In line 201-206, you selected a set of SNPs based on the criteria you set. What are these criteria based on? Please provide literature evidence to support the criteria you set are suitable.

3.    In table 5 and table 6, you demonstrated that genetic polymorphisms in the vWF and 260 ADAMTS13 genes were not associated with progressive severity of Covid-19 and but some of the SNPs were significantly associated with hemostatic parameters I found that apart from the hemostatic parameters you also collected the other clinical information of all the participants, like body mass index, respiratory rate and interleukin-6 which are all continuous variances. Are the SNPs you chose also associated with these parameters? You should provide the association results with other parameters to better support your conclusion.

4.    The participants were predominantly males, is the conclusion you claimed related to gender? Is the SNPs you selected demonstrating the same association trend in only the male participants?

Reviewer 3 Report

Majors:

-What is the date of the patient recruitment period? It is possible that some individuals might have been vaccinated. In the text, it is not clear whether COVID-19 patients were immunized. Whether they are positive for SARS-CoV-2 IgG should be specified.

-Unclear recruitment principles. It is unclear how healthy controls were recruited.

-In Table 2, this is more of a biological characterization than a clinical one...several clinical patient data are missing, including comorbidity, medication, gender, ethnicity, and number of deaths.

-Small study population. No medical history or medical treatment provided. All these factors could influence coagulopathy, making confounding and selection effects likely.

Reviewer 4 Report

In this study, the authors investigated a set of SNPs in the vWF (rs216311, rs216321, rs1063856, rs1800378, rs1800383) and ADAMTS13 genes (rs2301612, rs28729234, rs34024143) and their relation to COVID-19 severity. Although the topic is interesting, I have several concerns that should be addressed, as follows:

1-     Abstract: In line 21, the authors describe the study design as "case-control." I think it is a cross-sectional study.

2-     Introduction: Please provide a more comprehensive literature review on vWF and ADAMTS13-encoding genes and various SNPs that have been studied in relation to thrombosis and hypercoagulability. I recommend transferring paragraphs from the results (lines 208–226).

3-     Materials and Methods:

-        Participants: The diagnosis of COVID-19 should be confirmed via nasopharyngeal swab testing by PCR.

-        Please provide an ethical statement (ethics approval and informed consent).

-        The study included only 72 patients, which is a relatively small number, and further subgrouping would decrease the number of cases in each group, which may affect the reliability of the results. Please add a justification for the sample size used in this study (sample size calculation).

4-     Results:

-        There are many factors that affect the risk of COVID-19 severity, such as age, gender, co-morbidities, etc. Please use these data for the adjustment of the risk ratio for each genotype in Table 5.

-        It is important to measure ADAMTS13 expression, as well as vWF activity and Ag, and to correlate their levels among the studied SNPs.

-        Please define haplotypes in relation to severity as well as vWF and ADAMTS13 levels.

5-     Discussion:

-        Please acknowledge the limitations of this study, including the small sample size and lack of ADAMTS13 and vWF level measurement.

-        Please revise the conclusion to reflect the study findings in addition to the recommendations.

6-     There are several structural, grammatical, punctuation, and capitalization errors that require heavy English editing.

Reviewer 5 Report

The current study entitled “Missense variants of von Willebrand factor in the background of Covid-19 associated coagulopathy” has investigated whether common genetic variations of vWF and ADAMTS13 gene were linked to Covid-19 severity and hemostatic measures. SNPs in the vWF (rs216311, rs216321, rs1063856, rs1800378, rs1800383) and ADAMTS13 genes (rs2301612, rs28729234, rs34024143) were genotyped. Studies showed no polymorphism-disease severity correlation. However, analysis of variance (ANOVA) revealed associations with clinical parameters like INR, fibrinogen levels, red blood cell count, haemoglobin and creatinine levels. Importantly, in silico protein structural study revealed that these missense mutations had broad conformational modifications that may impact vWF stability and plasma levels. The findings suggest that missense vWF variants might modulate the thrombotic risk in Covid-19. I would recommend this manuscript with some improvements and queries.

·      As stated in the abstract, “Case-control analysis revealed no association of any polymorphism with disease severity,” but results only show disease cohorts analysis. 

·      The introduction needs a brief explanation of the rationale of SNPs selection for this study.

·      In the abstract, the Authors need to add the statement about the effects of ADAMTS13 genes SNPs (rs2301612, rs28729234, rs34024143) on the clinical parameters of the disease.

·      Methods do not mention how many total participants’ SNPs are genotyped. 

·      Page 4, line 168, this heading (2.6. Statistical analysis) needs to change accordingly and should be before 2.5 Statistical analysis.

Round 2

Reviewer 1 Report

This study has a huge problem in that the sample size is so small with only 72 individuals that the results cannot be trusted and are not considered worthy of a publication. Many authors have pointed out in the past that the minimum sample size for any genetic association study is 250. Results from studies smaller than that are often false positives or false negatives and cannot be trusted, since the results cannot be replicated in any subsequent study. Most studies now have thousands if not millions of specimens to obtain stable estimates of the genetic effects under study. We must continue to make sure that published studies meet the minimum sample size threshold, otherwise we are making the “reproducibility crisis of science” worse. I can understand the reasons that the authors give for the small size of their study. These reasons, do not, however, justify the publication of this study. The authors should take this as a learning experience and not try to force the publication of a study that will not contribute any useful knowledge to the field.

Reviewer 3 Report

The authors have done an admirable job of addressing all the concerns of the reviewers

Reviewer 4 Report

The authors have adequately addressed all my concerns.